# Meso-scale Simulation of Typhoon-Generated Storm Surge: Methodology and Shanghai Case Study

Shuyun Dong[1], Wayne J. Stephenson[1], Sarah Wakes[2], Zhongyuan Chen[3], Jianzhong Ge[3]

[1]School of Geography, University of Otago, Dunedin, 9016, New Zealand
[2]Department of Mathematics and Statistics, University of Otago, Dunedin, 9016, New Zealand
[3]State Key Laboratory for Estuarine and Coastal Research, East China Normal University, Shanghai, 200062, China

*Correspondence to*: Wayne Stephenson (wayne.stephenson@otago.ac.nz) +64 3479 8776

**Abstract.** The increasing vulnerability of coastal mega-cities to storm surge inundation means both infrastructure and populations are subject to significant threat. Planning for further urban development should include consideration of the changing circumstances in coastal cities to ensure a sustainable future. A sustainable urban plan relies on sound preparedness and prediction of future climate change and multiple natural hazards. In light of these needs for urban planning, this paper develops a general method to simulate typhoon-generated storm surge at the meso-scale (1 - 100 km in length). Meso-scale simulation provides a general approach with reasonable accuracy that could be implemented for planning purposes, while having a relatively low computation resource requirement. The case study of Shanghai was chosen to implement this method. The meso-scale simulations of two historical typhoons not only provides realistic typhoon storm-surge inundation results at the city level, but is also suitable for implementing a large amount of simulations for future scenario studies. The method will be generally applicable to all coastal cities around the world to examine the effect of future climate change on typhoon-generated storm surge, even when historical observation data is inadequate or not available.

**Keywords** Storm surge, Typhoon, meso-scale, simulation; Shanghai

## 1 Introduction

Rapid urban expansion in coastal mega-cities (cities with populations over 10 million) leads to increased land demand and vulnerability to hazards for significant numbers of people who are economically and socially disadvantaged. It is necessary to be well prepared and plan to ensure a sustainable future for these cities (Jiang et al. 2001; Timmerman and White 1997; Yeung 2001). Typhoon-generated storm surge is a major hazard for many coastal cities and leads to significant economic losses. Considering the ongoing coastal development and population growth in coastal mega-cities, preparedness and urban planning play a critical role in coastal management and hazard mitigation. Therefore, the increasing vulnerability of coastal mega-cities to storm surge inundation needs be assessed to improve the resilience of these cities (Aerts et al. 2014; Woodruff et al. 2013). Many integration models for typhoons and storm surge have been developed and applied in past studies to simulate regional storm surge inundation and analyse its impact (Choi et al. 2003; Davis et al. 2010; Dietrich et al. 2011b; Elsaesser et al. 2010; Flather et al. 1998; Jakobsen and Madsen 2004; Lowe et al. 2001; Westerink et al. 2008; Zhang et al. 2008; Zheng et al. 2013). In order to achieve more accurate results, high resolution mesh and data are usually employed in these models, which requires a large amount of computing time, and the application of such models are limited to small regions. As suggested by Aerts et al. (2014), existing hydrological models for developing inundation scenarios usually need to be adjusted for application at the regional level. A high-resolution storm surge model could therefore be too time consuming to be used for planning purposes when a large number of simulations need to be undertaken to gain a better knowledge of storm surge inundation. Ogie at al. (2019) argue that there is a need for less data rich approaches to flood model of coastal mega-cities where there is often a paucity of data. The purpose of this paper is to develop a less resource intense simulation method for typhoon storm surge inundation at a city scale and to implement this method using Shanghai as a case study. The approach developed was to conduct numerical simulations of typhoons and their associated storm surge at a meso-scale (1 - 100 km in length), which can then be utilized to compute flooding scenarios.

## 2 Previous Work

There is a large amount of previous work on the storm surge modelling. Regardless of the models used, previous studies can be divided into the three types based on scale of modelling, namely large-, meso-, and small-scale. For large scale storm surge studies, they usually concentrate on simulating storm surge at the national level (>100 km in length). For example, Lowe et al. (2001) developed a storm surge with 35 km resolution for the North west European continental shelf region, and then analysed the effects of climate change using a regional climate model. Fritz et al. (2010) simulated the storm surge occurring in the

Arabian Sea with a spatial resolution range of 1 – 80 km. Haigh et al. (2014) utilized a high-resolution hydrodynamic model to estimate extreme water level exceedance probabilities for the Australian continental shelf region. Due to the high risk of storm surge, there have been studies conducted for the Louisiana coast, USA (Butler et al. 2012; Sheng et al. 2010; Wamsley et al. 2009; Westerink et al. 2008) and the Gulf of Mexico area (Dietrich et al. 2012; Dietrich et al. 2011a; Dietrich et al. 2011b). Cheung et al. (2003) analysed the emergency plan for Hawaii based on storm surge simulations. These large-scale storm surge studies normally apply a large spatial resolution, 50 – 100 km on average, to allow the simulation to be run smoothly in a large-scale area. It is inevitable that at such large spatial resolution small variations in terms of storm surge level at regional level is lost, making it a less suitable type of model for studying the impact of inundation at a city scale (typically 20-80 km length of coast).

Meso-scale storm surge modelling typically focuses on a scale of 1 - 100 km in length. Peng et al. (2004) utilized an integrated storm surge and inundation models to simulate storm surge inundation in the Croatan–Albemarle–Pamlico Estuary in US. Shepard et al. (2012) demonstrated a method to assessing community vulnerability of the southern shores of Long Island, New York to storm surge. For small-scale storm surge studies, the focus is at a regional level (1 – 1000 m in length). Taking the study of Funakoshi et al. (2008) as an example, a fine small-scale storm surge model was developed covering the St. Johns River Basin in USA. Xie et al. (2008) developed storm surge modelling to simulate corresponding inundation. Frazier et al. (2010) examined the socioeconomic vulnerability to storm surge in Sarasota County, Florida, USA. Small-scale storm surge studies normally focus on the effect of storm surge at a local level and is commonly used to provide advice for small-scale planning and emergency management.

There are a number of storm surge studies conducted in China, and hydrological models for storm surge simulation have been developed. However, these are either at a large or small-scale, which may lead to a loss of spatial resolution in the simulated storm surge results or in huge costs in computation time. Most of these studies emphasized the significance of numerical modelling of storm surge and risk analysis either for the coastline on a large spatial scale or for the local coastal area with fine resolution simulation. For example, Zheng (2010) developed a numerical model to simulate storm surge under the effects of tide and wind wave for the coast of China. In 2011, Tan et al. (2011) assessed the vulnerability of coast cities in China to storm surge using an indicator system. Yin (2011) also assessed the China coastal area's risk to typhoon storm surge based on the simulated results from large scale storm surge model and a proposed indicator system. Other studies placed emphasis on the analyse of storm surge at small regional scale areas along the Chinese coast (Xie 2010; Xie et al. 2010; Ye 2011; Zhang et al. 2006).

This study, therefore, utilizes a meso-scale (between large and regional scales) approach for inundation vulnerability to typhoon storm surge to improve knowledge of inundation vulnerability and to guide future vulnerability mitigation strategies. Moreover, a large number of simulations are involved in planning. Therefore, in order to fit this purpose, a meso-scale study for Shanghai as a whole is utilized, these filling the gap between the small and large-scales of previous studies. In addition, this meso-scale simulation aims to provide a general approach that could be easily implemented for other coastal cities and has much lower requirements for computation time and data than previous approaches.

**3 Data and Methods**

The objective of this general methodology for simulating typhoon storm surge inundation is to develop an adaptable procedure that allows numerical simulations to be carried out easily in coastal cities around the world. Firstly, the data required in the typhoon and storm surge simulations was assembled, including the observation data from typhoons, tidal constituents,

topography, and land use data. Two historical typhoons were selected to develop typhoon profiles and then wind and pressure fields were calculated to drive the hydrodynamic storm surge model. Typhoon wind and pressure field was calculated based on historical typhoon profiles. Moreover, tidal observed data was collected to validate the hydrodynamic models in the next step. The next step was to implement a storm surge model to simulate the generation and propagation of typhoon-driven storm surge model at coastal and regional scales. The historical wind and pressure fields are inputs into the coastal hydrodynamic model along with the tidal constituents as key driving factors to simulate the initial current and wind-induced surge at coastal scale. Then, considering river discharge and coastal protection works, storm surge is simulated using the regional hydrodynamic model with a fine spatial resolution unstructured mesh. Lastly, the flood depth can be extracted from the simulation results in regional hydrodynamic model and overlaid onto the urban digital elevation model, where the flood depth and its spatial extent are displayed on a two-dimension flood map. The proposed method is explained in following sections.

**3.1 Assembling Data**

An accurate wind and pressure field has been identified as having an important role in storm surge modelling (Bode and Hardy 1997). In order to provide wind and pressure field to drive storm surge in hydrodynamic model, historical typhoon data needs to be collected from records. There are various types of typhoon data, such as best track data, observed data, and satellite data. Typhoon wind and pressure field are calculated in this framework by applying the parametric model built in MIKE 21 Cyclone Wind Generation tool. Typhoon data required in the simulation then are the typhoon track, the central and neutral air pressure, and the maximum wind speed. This data can usually be found in best track data published by meteorological agencies (Ying et al. 2014) or satellite reanalysis database (Simmons et al. 2007). The development and optimization process of typhoon wind modelling is described in Section 3.3. To pre-process the data for the subsequent modelling, all the historical topography and meteorological data was digitized into appropriate formats, including bathymetry, urban digital elevation model, land use map, and coastal engineering features. In this step, tide constituents are prepared in the format that is required in storm surge modelling.

**3.2 Developing a Storm Surge Coupled Model**

Water propagation at the coast is significantly sensitive to surface wind forcing and astronomic tides, especially during typhoon events. As suggested by Huang et al. (2010), wave-induced forces on storm surge are incremental, so there is no need to utilise an independent wave model. Therefore, in this study, a coupled model will be built to simulate storm surge. In order to provide accurate wind and pressure fields and tide influence for the coastal and regional circulation, a two-domain, Typhoon Storm Surge Model was set up, covering the coastal and regional geographic scales. In this method, MIKE 21 was chosen to simulate typhoon-generated storm surge with consideration of river discharge and coastal protection works. As commercial software, MIKE 21 has broad applicability and a low requirement of specialized knowledge. In general, a hydrodynamic model for a coastal area will be set up and calibrated against observed tide data. Then the coastal hydrodynamic storm surge model will be utilized to calculate the corresponding distribution of the wave field under the influence of a historical typhoon wind and pressure field. On this basis, a regional storm surge model can be built for shallow water to consider wave refraction, diffraction, and transformation in order to calculate storm surge in the area of interest. After calibration against measured historical data of storm surge, this model can be applied to project the impact of future storm surge for the study area.

3.2.1 Grid Model and Resolution

In order to precisely simulate storm surge in any coastal area, a fine grid model with appropriate resolution should be constructed for the coastal terrain and topography. The grid greatly affects the generation, propagation and reflection of the wave, and bottom friction. However, a very fine grid resolution causes significant increases in the computing time and resource. Thus, a balance between accuracy of numerical simulation and the computing requirement should be achieved in the model.

The resolution of the unstructured mesh applied in the coastal hydrodynamic model is recommended to be set in a range of 1
km at the coastal zone to 10 km at the open ocean boundary (Fig 3.1). For the regional hydrodynamic model, the resolution
can be more precise with an average of 300 m.
3.2.2 Coastal Hydrodynamic Model
Wind and pressure fields of the typhoons, together with astronomic tide and waves are the main factors of storm surge that
need to be taken into account in simulations (Savioli et al. 2003). Combining the statistical hydrological and meteorological
data, a coastal typhoon storm surge model is designed and developed using MIKE software to simulate historical storm surge
events, which in turn allows simulation of the hydraulics, waves and related phenomena in the coastal area. This coastal
hydrodynamic model with a flexible mesh is built up in the MIKE 21 flow model to simulate wind-generated waves and current
conditions with respect to pre-processed tide, wind and pressures fields. This coastal typhoon storm surge model was first
calibrated under normal circumstances to fit no storm tidal conditions firstly, then run for historical typhoon storm surge events
to ensure the reliability of simulations. First, the coupled model was only run to compute tide parameters during the three days
before the typhoon for the entire region for the purpose of calibration. Then the model was run to simulate historical typhoon
events and the simulation calibrated with observed data of surge elevation. In addition, computed data of wind speed and
direction were calibrated against satellite data or local measured data.
3.2.3 Regional Hydrodynamic Model
Based on the computed data from the coastal hydrodynamic model, a regional model was developed to simulate the movement
of typhoon-induced surge for a relatively small regional area. Then this regional model was run for different scenarios, to
project the effects of global climate change and land subsidence on the regional storm surge level. This regional hydrodynamic
model can provide predicted results under various scenarios for decision making, hazard mitigation and emergency evacuation
planning. By analysing various future scenarios, a better understanding of coastal vulnerability can be reached, then appropriate
preparedness and mitigation planning can be made.
3.2.4 Major Model Parameters
The hydrodynamic module in MIKE 21 Flow Model was employed in this study to implement the coastal and regional
hydrodynamic models. A number of model parameters need to be set ahead of running simulations, so these are now described.
The horizontal eddy viscosity is specified as a constant of 0.8 is taken from Smagorinski, (1963) and used in the SC-TSSM
(Shanghai Coastal Typhoon Storm Surge Model). The effect of different shapes of sea walls in the storm surge model is minor,
therefore the shape of the sea wall was assumed to be trapezoidal. In our case study below, the height of the sea wall along the
Shanghai coastline is 6.37 m relative to mean sea level and this value is used in the model. Boundary conditions in the open
sea are driven by the astronomical tide. In this study, the tide profile before and during the typhoon period was computed by
the Global Tide Model in MIKE. TOPEX/POSEIDON altimetry data have been employed in the Global Tide Model with a
spatial resolution of 0.25 ° * 0.25 °. The output data of boundary condition files have a 1-hour interval. Other parameters
configured in the coastal and regional hydrodynamic models are listed in Table 1.
**3.3 Storm Surge Inundation Modelling**
For large-scale and meso-scale studies, storm surge inundation mapping can be conducted to predict the inundation depth and
spatial extent. The approach to inundation mapping can also be utilized for the purpose of further planning which aims to
predict the distribution of storm surge inundation, especially in land reclamation planning. Based on the typhoon storm surge
simulation results from the regional hydrodynamic model, inundation maps are constructed using ArcGIS. Flood maps drawn
in ArcGIS provide graphic information with which to analyse the differences in inundation depth across the city.

### 3.4 Optimizing Process in Wind Field Simulation by MIKE Software

In order to analyse the storm surge caused by typhoons, a precise simulation is closely bound to the accuracy of wind and
pressure field specification. It is therefore of considerable significance that a specific, accurate and representative typhoon
field is input into the typhoon model. In this study, the wind and pressure field of the typhoon was generated by the parametric
model in the MIKE 21 Cyclone Wind Generation tool. There are four parametric models built in this tool; Young and Sobey
(Young and Sobey 1981), Holland (Holland 1980), Holland for double vortex (Harper and Holland 1999), and Rankine (1872).
The Holland model has been chosen to simulate the typhoon wind field in the Shanghai case study because the adjustability of
the Holland parameter $B$ allows the model to be modified to fit existing data more realistically.
Most of the parameters in the Holland model can be collected from the typhoon best track dataset of the China Meteorological
Administration and the European Centre for Medium-Range Weather Forecasts (ECMWF) (Molteni et al. 1996). The best
track data were recorded every 6 hours, then the model will simulate the wind and pressure field at 1-hour interval. The
remaining two parameters, the radius of maximum wind $R_{mw}$ and parameter $B$, was calculated by Eq. (1) (Ge et al. 2013) and
Eq. (2) (Vickery et al. 2000) respectively.
$$R_{mw} = (7.5757576 \times 10^{-5}) \times P_c^2 - 0.50560606 \times P_c + 477.01515 \qquad \text{Eq. (1)}$$
$$B = 1.38 - 0.00184|P_c - P_n| + 0.00309R_{mw} \qquad \text{Eq. (2)}$$
where $P_c$ represents the pressure at the typhoon centre or central pressure, $P_n$ is the ambient pressure field or neutral pressure.
Although the computed results by the Holland model show that the model is in good agreement with the actual observation, a
relative error remains in the computation after typhoon landfall (Fig. 1). Compared to the observation data, the computed wind
speeds fall rapidly after the typhoon made landfall. In order to improve the quality of typhoon simulated results, a commonly
applied approach is to blend computed wind speeds results with satellite reanalysis database, such as global National Centres
for Environmental Prediction and National Centre for Atmospheric Research (NCEP/NCAR) Reanalysis data and ECMWF
reanalysis dataset (Dutta et al. 2003; Jia et al. 2011). The ECMWF reanalysis dataset has a better spatial resolution of 0.25 °
than NCEP/NCAR (2.5 °). Therefore, the ECMWF dataset was chosen here as the background wind field to achieve a more
precise result at the outer area of the radius of maximum wind.

(a)

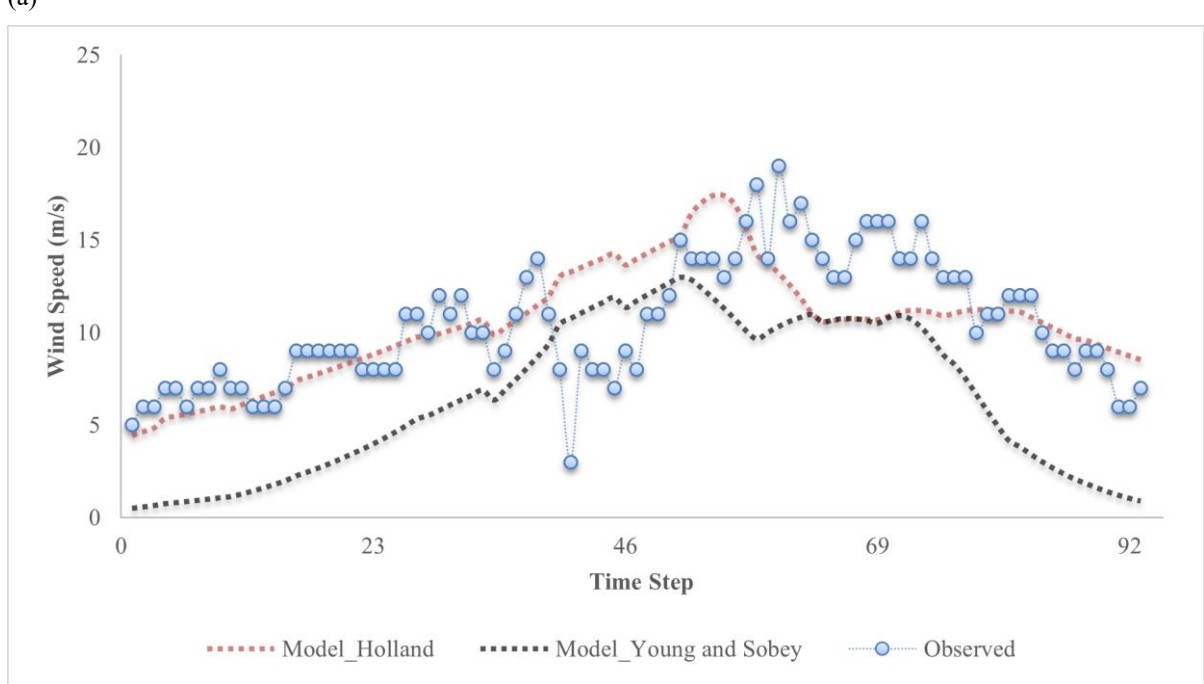

(b)

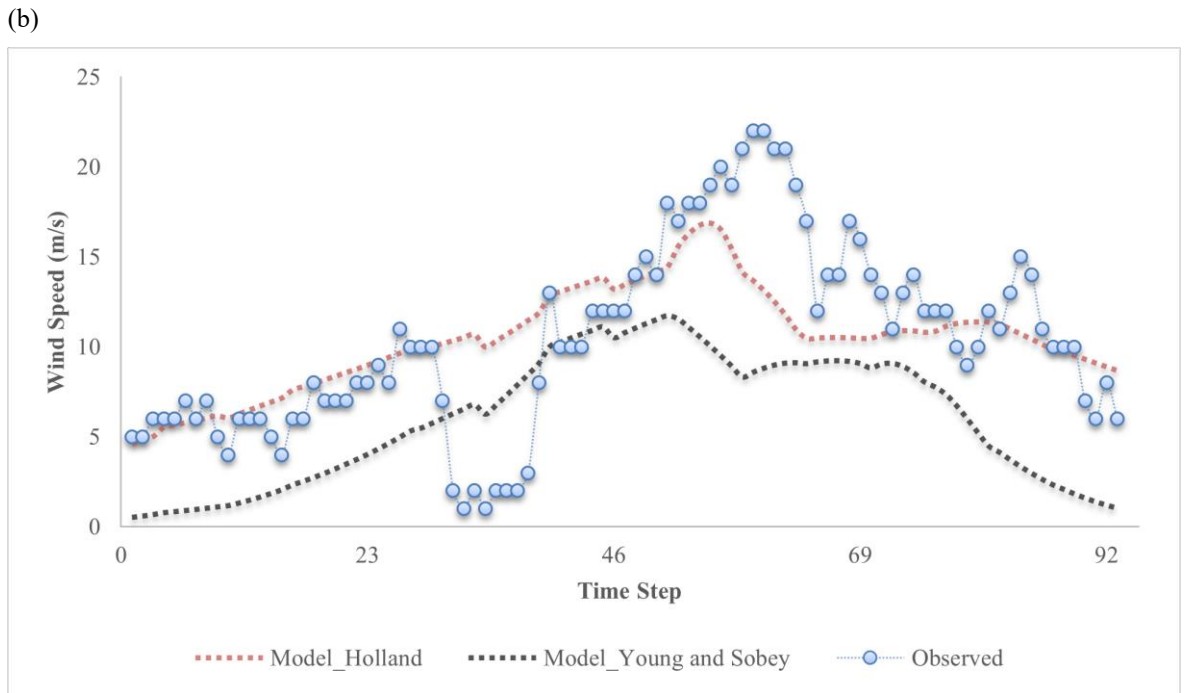

**Fig. 1** Model data comparison for (a) results at Tanxu station, and(b) results at Daji station. The blue points indicate the observation data, the red curve shows the simulated results following the Holland model, and the grey curve represents the results computed by the Young & Sobey model.

The ECMWF reanalysis dataset is a continually updating dataset with the finest resolution of a 0.25 ° * 0.25 ° grid mesh
presented by the European Centre for Medium-Range Weather Forecasts. It has been recording joint data from diverse,
advanced, operational, numerical models, representing the state of the Earth's atmosphere, incorporating observations and a
numerical weather prediction model four times daily since 1948 (Simmons et al. 2007). As a result of the assimilation of the
observational data, the recorded atmospheric circumstances in the ECMWF dataset can be regarded as providing a close
approximation of the state of the atmosphere (Molteni et al. 1996). Therefore, the ECMWF can provide a precise, nearly real
atmospheric background for adjusting the Holland model.

In order to integrate the strong points of the MIKE software and the ECMWF reanalysis dataset, the MIKE Software Development Kit (SDK) is used here to optimize the simulation results from the MIKE 21 Cyclone Wind Generation Tool. The wind speed $V(r)$ at a distance $r$ from the centre of the typhoon, can be given by Eq. (3):

$$V(r) = \begin{cases} V_{MIKE} & , r < R_{mw} \\ V_{ECMWF} & , r > R_{mw}, \\ aV_{MIKE} + (1-a)V_{ECMWF}, & r = R_{mw} \end{cases} \qquad \text{Eq. (3)}$$

where $V_{MIKE}$ is the wind speed calculated by the MIKE 21 Cyclone Wind Generation Tool, $V_{ECMWF}$ is the wind speed computed from the ECMWF interpolation results, and $a$ is the weight factor in order to smooth rough edges. An optimized coupled wind and pressure field can be generated by programming in the MIKE SDK based on Eq. (3). This produced a wind and pressure field that matched the actual typhoon event well.

**4 Case Studies in Shanghai**

Following the proposed framework for assessing inundation vulnerability to storm surge, a case study of Shanghai is used to examine the application of this proposed approach (Fig. 2). There were 16 major storm surge events in Shanghai from 1905 to 2000; five of them (in 1905, 1933, 1981, 1997, and 2000) have led to severe flooding and billions of Yuan in economic damage.

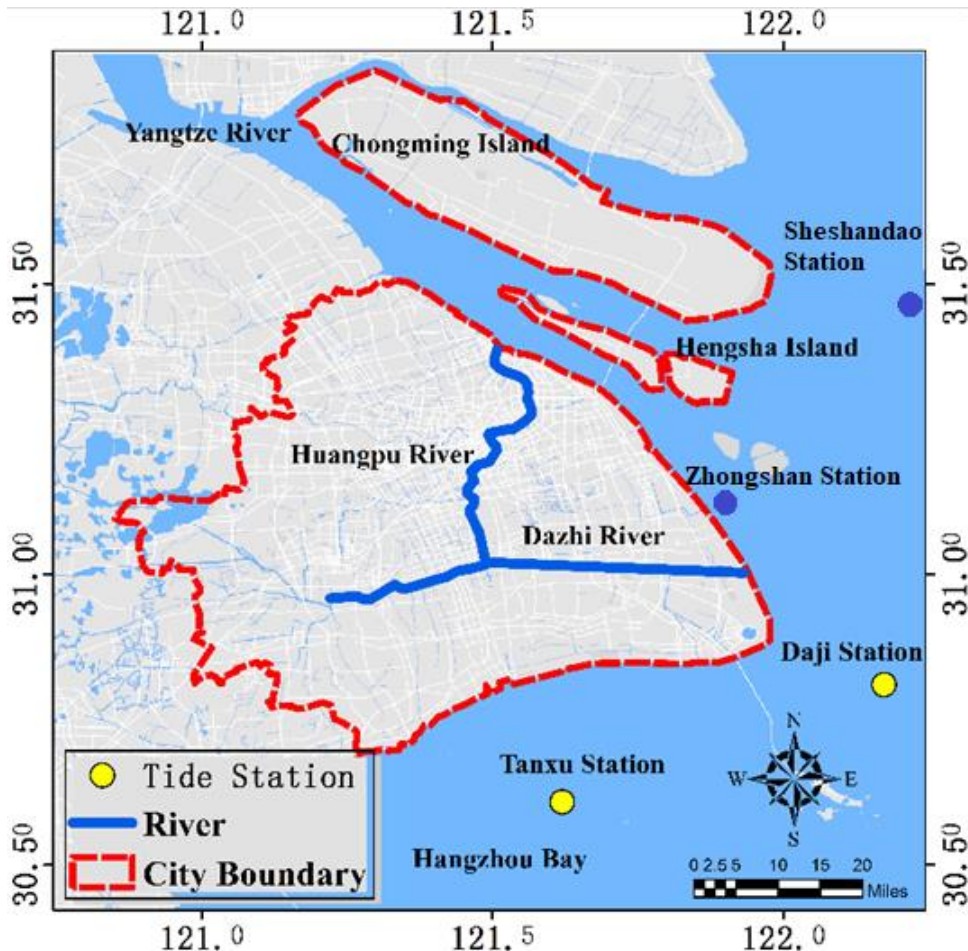

**Fig. 2** Local map of Shanghai with the red dash line indicating city political boundary and simulation area, while the blue line indicating the Huangpu and Dazhi Rivers. The yellow points represent two tide gauges used to calibrate the models, while the blue points represent tide gauges used for tide validation. Sources: Esri, DeLorme, HERE, USGS, Intermap, iPC, NRCAN, Esri Japan, METI, Esri China (Hong Kong), Esri (Thailand), MapmyIndia, Tomtom.

Along the Shanghai coast, land reclamation has grown substantially due to the increasing demand for land for further urban development, about 480 km$^2$ land was been claimed in Shanghai between 1954 and 1990 (Shanghai Nongken Chronicles Compilation Committee 2004). Reclaimed land can alleviate the pressure on land that results from the continuous growth of cities in the process of rapid expansion. Most of the newly reclaimed land has been used for agriculture and industry (Shanghai Municipal Planning and Land & Resources Administration 2010). However, such extensive reclamation activities require long-term, well-developed planning, otherwise there may be increased vulnerability and even catastrophic damage due to natural hazards.

Typhoon Winnie in August 1997 and Typhoon Wipha in September 2007, were chosen as case studies to simulate typhoon storm surge and assess the vulnerability to typhoon storm surge inundation of differing land use types under sea level rise and land subsidence scenarios. Both Winnie and Wipha were categorised as super typhoon by the China Meteorological Administration and caused serious storm surges in Shanghai. These two typhoons affected a wide-ranging area, so simulation results could provide more information on the vulnerability of different land use types under worse case scenarios. In addition, Winnie and Wipha represented typical turning track typhoon. They developed in the northern Pacific Ocean, and then tracked north-westward to China. After they passed across the East China Sea, they moved north-eastward. As with the majority of typhoon affecting Shanghai, although they did not make landfall directly at Shanghai, they generated high storm surge in Shanghai, 5.72 m during Winnie and 3.39 m during Wipha. In addition, the 10 years' interval between these two typhoons could allow the simulations to reveal how inundation vulnerability of different land use types to typhoon storm surge changed over time.

Typhoon Winnie (1997) was an especially large and devastating typhoon. After passing north of Taiwan, Winnie made landfall at the south-east of Shanghai in Wenling, Zhejiang province on 18 August 1997. Its centre was never closer than 400 km from Shanghai, however the storm surge caused by Winnie led to extraordinary levels of flooding. Winnie gave rise to 212 deaths, over 1 million people were displaced, and there are 4.1 billion yuan of economic losses (State Oceanic Administration 1989-2015). A resulting storm surge of up to 6.57 m was measured at Jinshanzui Station. After landfall, Winnie shifted from the northeast to northwest, giving rise to approximately 37 km of riverbank overflowing and 70 km of dike breaches (Zhu et al. 2002). A storm surge with a wave height of approximately 7.9 m developed in Zhejiang province, then this decreased to around 5.72 m as it approached the Shanghai area. Typhoon Wipha (2007) was another destructive typhoon which passed near Shanghai and landed in Cangnan, Zhejiang province on 19 September 2007. As a typical turning track typhoon, it passed to the west of Shanghai after making landfall to the south. Although the eye of Wipha did not pass near Shanghai, its outer strong wind and rain bands resulted in severe flooding in Shanghai. Although the recorded highest water level in Shanghai was only 3.39 m during this typhoon on 19 September 2007, there were 128 roads flooded and over 1 million Yuan (2007) of direct losses were caused in Shanghai. In addition, almost 300,000 people had to be evacuated by the Shanghai government (State Oceanic Administration 1989-2015).

**4.1 Required Data and Processing**

Data regarding topography and meteorological data from Shanghai in both 1997 and 2007 were collected and processed before modelling. Assimilated wind data were required using best track data from the China Meteorological Administration Tropical Cyclone Data Centre and ECMWF Global Reanalysis Products with a resolution of 0.25 °. Both of these two datasets have a 6-hour interval, therefore integrate well with each other in the typhoon model to improve the accuracy of simulated results.

The computational models in this study for storm surge simulation employ an unstructured mesh spacing of 1 km in the regional area and 100 km for the open sea area. The topographical data applied in the urban area to generate the flexible mesh was provided by the East China Normal University. The topographical data was extracted from the digital elevation model of

Shanghai with a 5 m spatial resolution. Bathymetry was taken from the ETOPO1 Global Relief Model downloaded from
NOAA with a grid resolution of 1 arc-minute in the open sea area, while data provided by the East China Normal University
with a spatial resolution of 1 km were adopted to improve the accuracy of the bathymetry data near shore (Fig. 3).

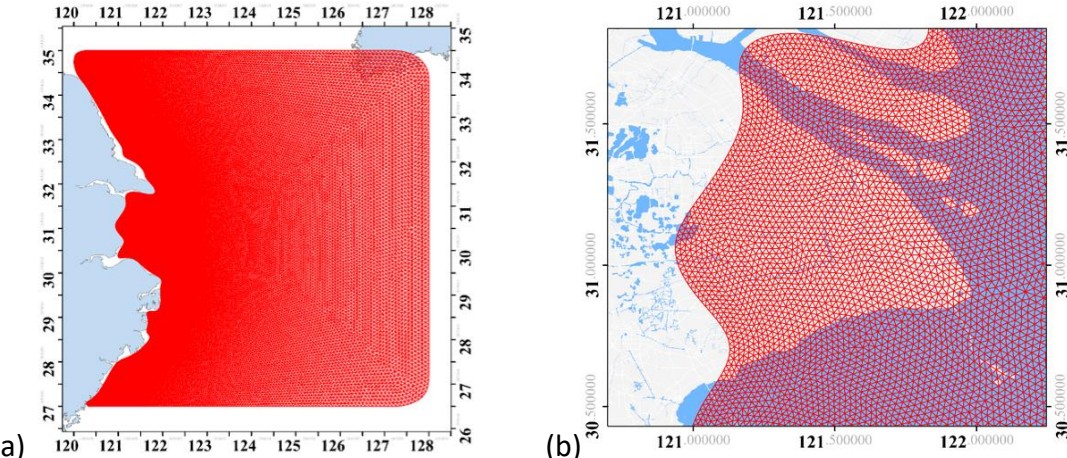


**Fig. 3** Shanghai Coastal Storm Surge Model with the resolution varying from 10 – 100 km. (a) shows the unstructured mesh
with the differing resolution, ranging from 10 – 100 km. (b) provides an enlarged image of the mesh around Shanghai. Sources:
Esri, DeLorme, HERE, USGS, Intermap, iPC, NRCAN, Esri Japan, METI, Esri China (Hong Kong), Esri (Thailand),
MapmyIndia, Tomtom

Four gauge stations were utilized here to validate and calibrate the simulated results from the typhoon and storm surge models.
For the purpose of model validation, the SC-TSSM was run for a period of one month before both historical typhoon events.
In these simulations, the effect of wind forcing was not taken into account in order to compare the plain model results with
actual data at the Sheshandao and Zhongjun Stations. Since observed tide levels at these gauge stations are not available during
the selected typhoon events. In order to validate the coastal storm surge model, the tide level extracted from a tide table was
adopted. The comparison between extracted data and simulated data is shown in Figure 4. It shows that the SC-TSSM has
produced good simulations for tide propagation in the coastal domain. From Figure 4, the simulations show a good agreement
with the extracted data from the two gauge stations. At Sheshandao and Zhongjun Stations, overall errors of 3.30 % and 0.52
% occurring during Winnie and Wipha respectively. Computed wind and storm surge results from numerical models have
been calibrated based on observation data at the two gauge stations off the coast of Shanghai, at Daji and Tanxu Stations (Fig.

275 2).

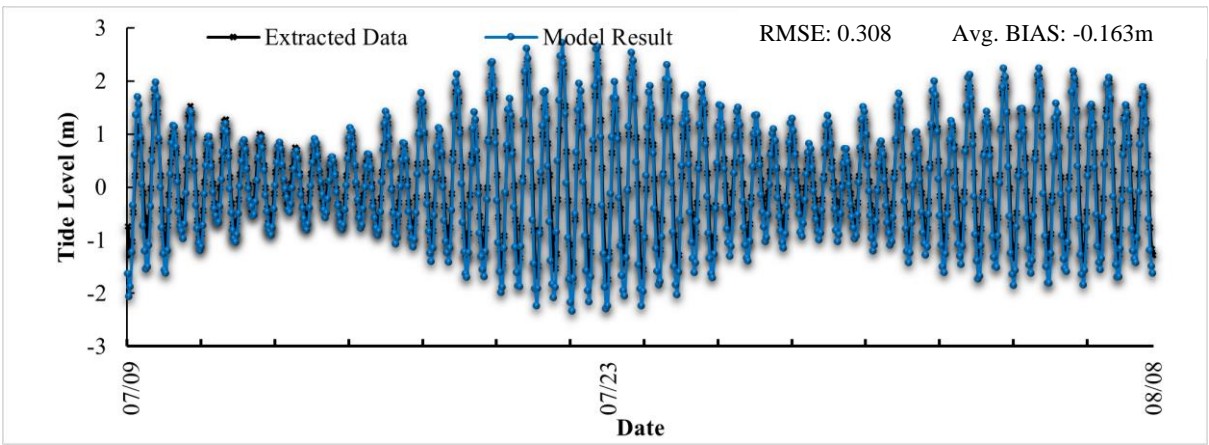

(a) one month before Typhoon Winnie at Sheshandao Station

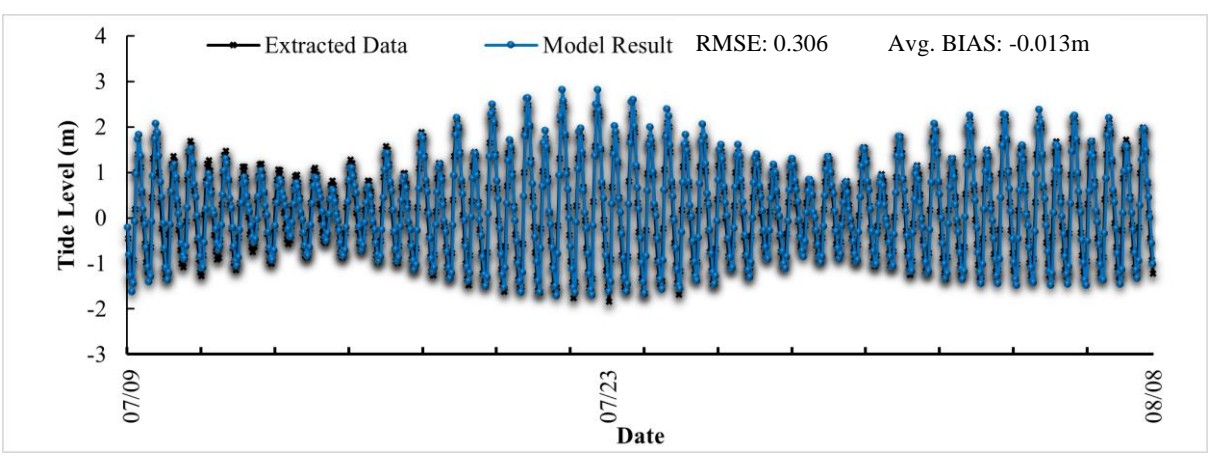

(b) one month before Typhoon Winnie at Zhongjun Station

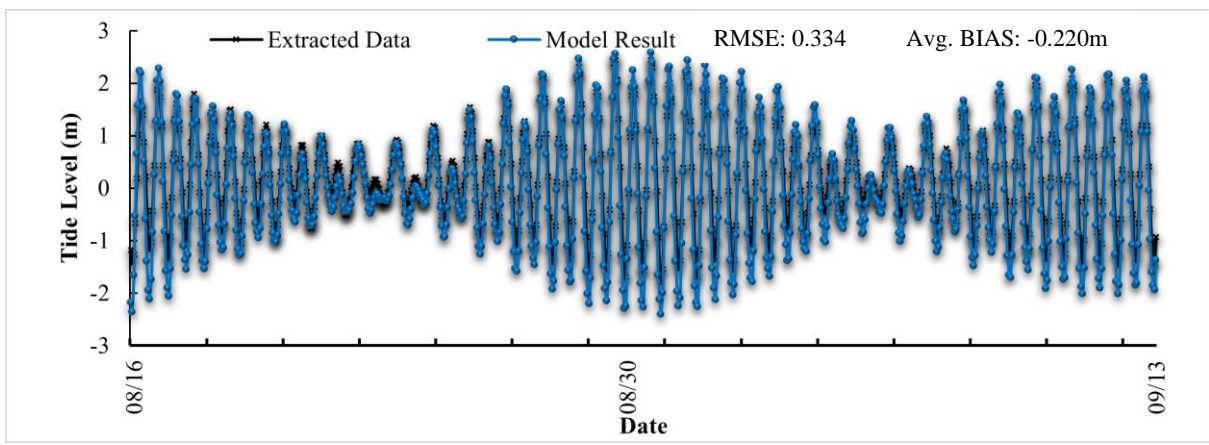

(c) one month before Typhoon Wipha at Sheshandao Station

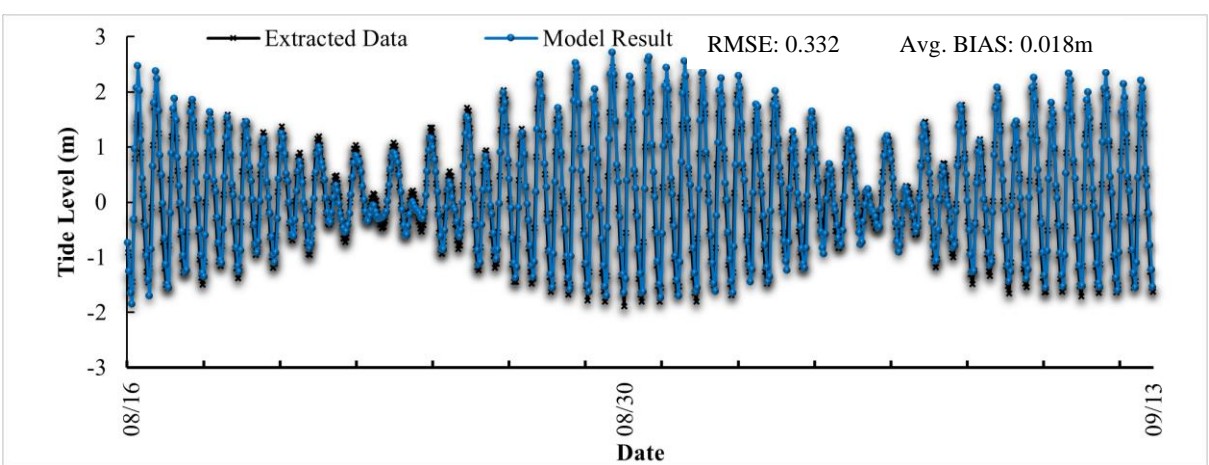

(d) one month before Typhoon Wipha at Zhongjun Station

**Figure 4** Time series of tide level (unit: m) at the tide gauge stations (Sheshandao and Zhongjun Station) presented in Figure 2. (a) and (b) present tide level at Sheshandao and Zhongjun Station from 8 July 1997 to 8 August 1997 before Typhoon Winnie, while (c) and (d) present tide level at Sheshandao and Zhongjun Station from 15 August 2007 to 15 September 2007 before Typhoon Wipha. The black line indicates the extracted data, while the computed results from the SC-TSSM are shown with the blue line.

## 4.2 Typhoon Modelling

In this study, the impacts of typhoons are derived from the wind and pressure fields using the MIKE 21 Cyclone Wind Generation tool. In order to improve the accuracy of the simulated results, the reanalysis dataset from ECMWF has been applied in MIKE SDK. Details are given in the following sections regarding the setup, calibration, and computed results of Typhoon Winnie and Wipha.

The typhoon model produces an output with a 1-hour interval, including the air pressure, and U and V components of wind speed. Afterwards, the simulated results have been passed to the storm surge model to generate wind-induced waves. The dataset used to initialize and, subsequently, simulate wind and pressure field in MIKE 21 was extracted from the best track data published by the CMA Tropical Cyclone Data Centre. The data for model optimization in MIKE SDK were a ECMWF reanalysis dataset with 6-hour intervals and a resolution of 0.25 ° * 0.25 °. In this study, the wind and pressure fields were generated with the parametric model of Holland's wind field profile for the area between 30 – 35° N, 120 – 130° E. ETOPO1 data and local measured data were employed to develop a topographical profile of the entire coastal domain.

The Holland parameter $B$ was set using Eq 2. A geostrophic correcting parameter can be implemented as a constant or varied according to the wind speed at different places. In order to correct the asymmetrical forward movement of a tropical cyclone, a correction factor $\delta_{fm}$ and the maximum angle of cyclone movement are introduced into the model to adjust the wind profile. In the case of Shanghai, $\delta_{fm}$ was set to 1 as recommended in the MIKE 21 user manual. The maximum angle was set to 115 ° and 150 ° as the maximum angles of Winnie's and Wipha's movements, respectively. Observed data from two wind gauge stations (Daji and Tanxu) have been used to calibrate the typhoon model. Results were outputted from the Holland model at 1 hr intervals and compared against observed data (Figs. 5 and 6). For both typhoons, calibration results of wind speed show that the simulation agrees well with measured data before each typhoon made landfall at Daji and Tanxu Stations. After the landfall, the simulation shows a 17.9 % and a 14.4 % mean absolute percentage error against observed data. The reason for this large increase in the error after landfall is mainly the long distance between the track of both typhoons and the wind gauge

stations. In addition, previous studies suggested significant fluctuations during typhoon events may be related to regional wind fields rather than the wind field driven by the typhoon (Zhu et al. 2002). Thus, the simulations around these two gauge stations failed to capture such fluctuations in wind speed. Although the simulated data cannot reflect minor changes in wind direction at shorter time intervals, they still have the same trend as do the observed data (Fig. 5). After calibration of the model, the computed results have been passed to MIKE SDK, and integrated with ECMW.

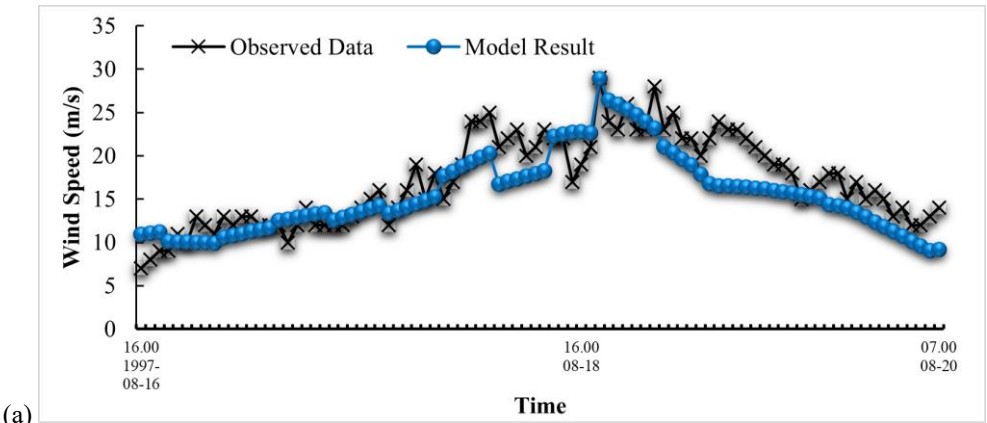

(a)

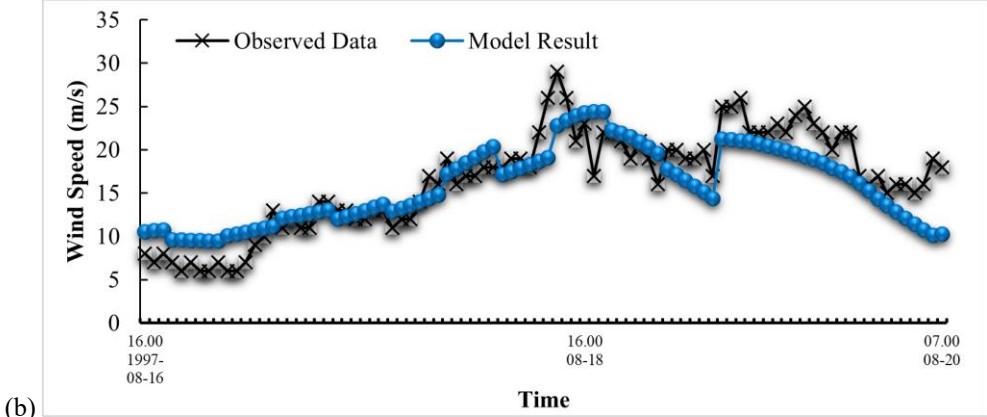

(b)

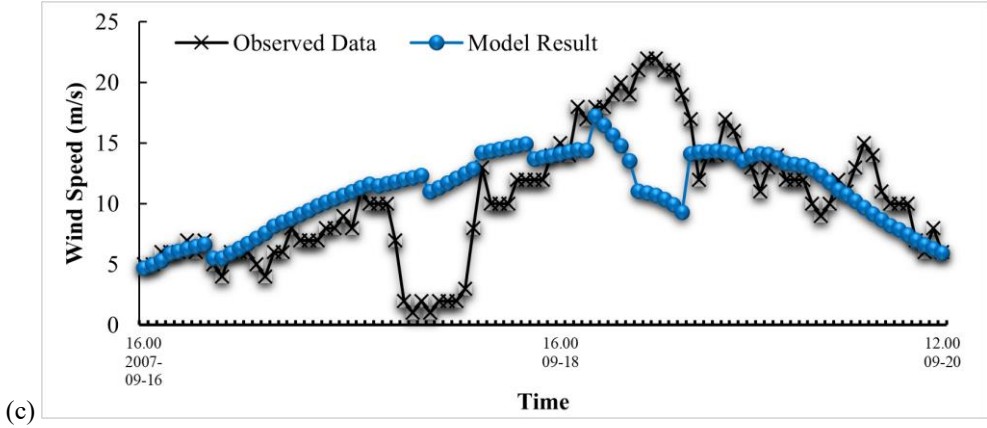

(c)

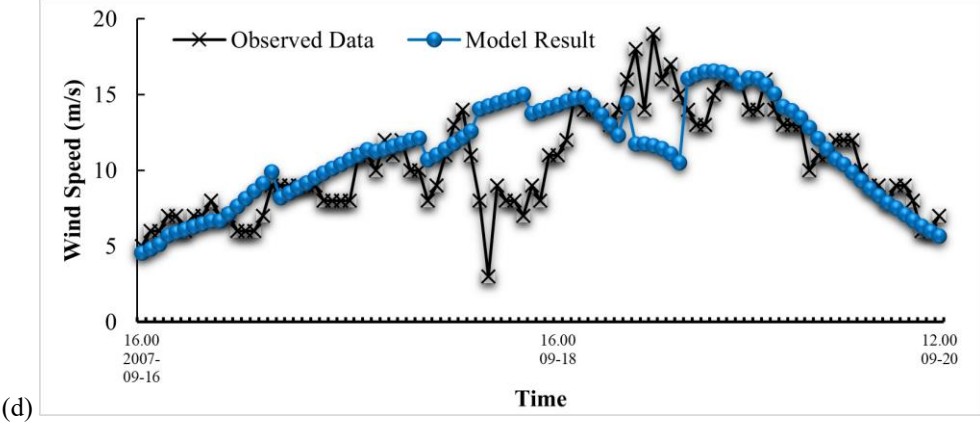

(d)

Fig. 5 Data comparison between computed wind speed (m/s) and observed data at the two wind gauge stations in Shanghai during Typhoon Winnie and Wipha. The blue line presents the simulated results from the typhoon model, while the black line indicates the measured data at the gauge stations. (a) and (b) Winnie. (c) and (d) Wipha at Daji and Tanxu Stations

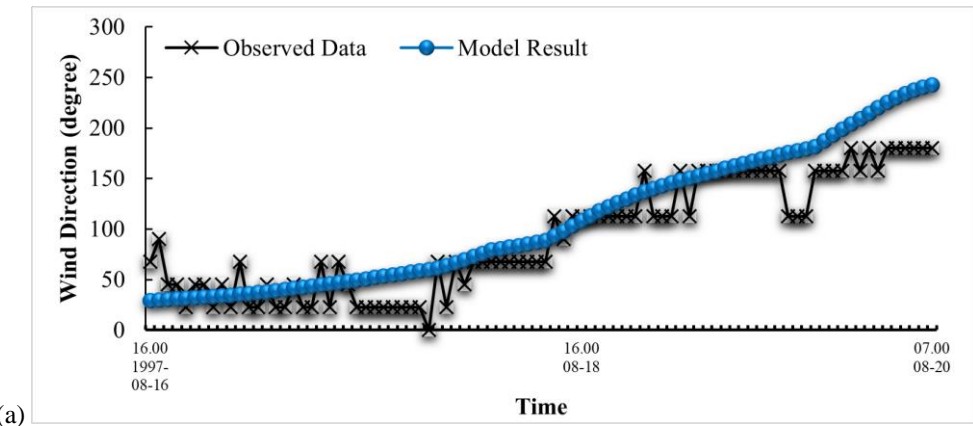

(a)

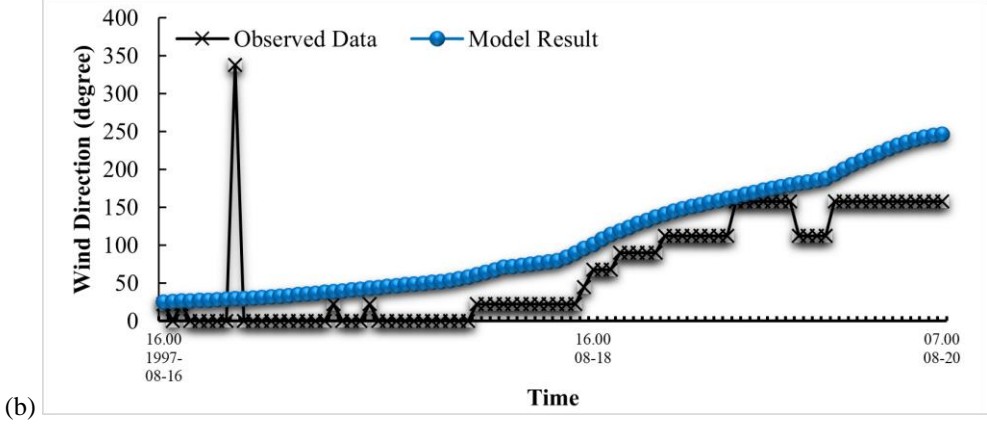

(b)

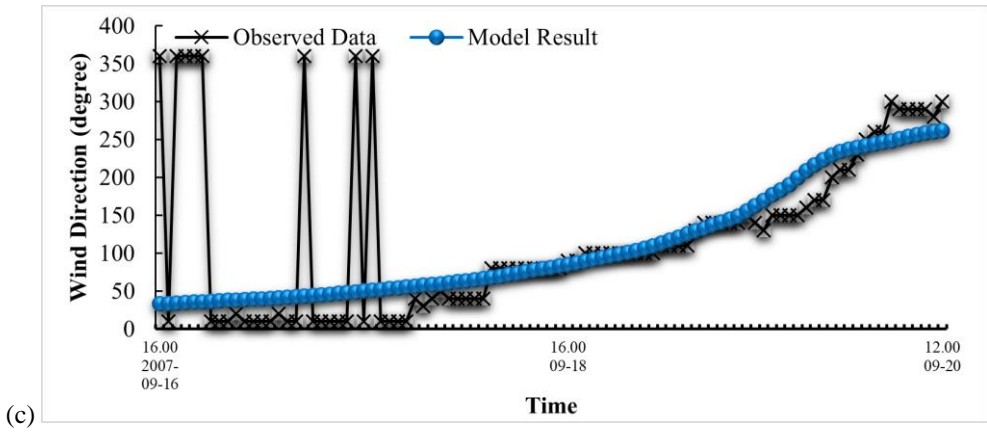

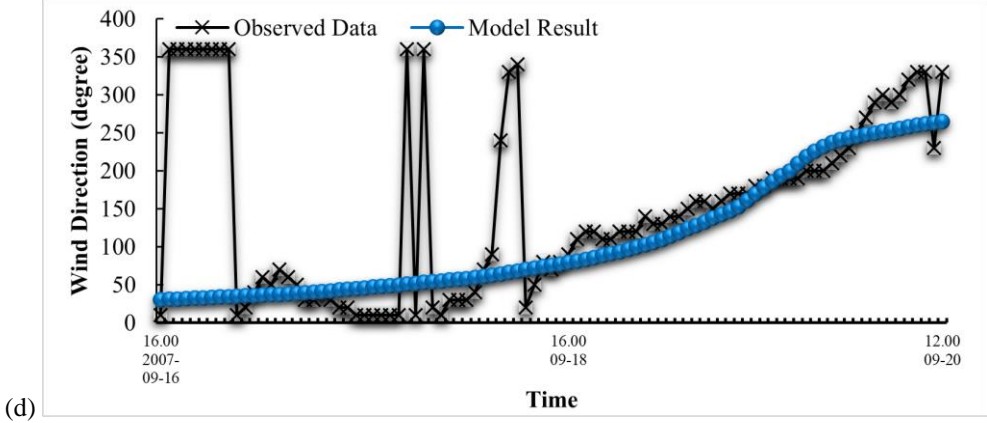

**Fig. 6** Data comparison between computed wind direction (degree) and observed data at the two wind gauge stations in
Shanghai during Typhoon Winnie and Wipha. The blue line presents the simulated results from the typhoon model, while the
black line indicates the measured data at the gauge stations. (a) and (b) Winnie. (c) and (d) Wipha at Daji and Tanxu Stations
**4.3 Results of the Shanghai Coastal Typhoon Storm Surge Model**
The typhoon simulation results were used as input into a storm surge model to provide the wind profile. In order to simulate a
typhoon-generated storm surge at coastal and regional scales, a Shanghai Coastal Typhoon Storm Surge Model (SC-TSSM)
was developed here. In this section, the configuration, validation, and calibration of the SC-TSSM will be described in detail,
and the simulated results of a storm surge caused by two selected typhoons will be discussed accordingly. SC-TSSM covers
the Shanghai coastal area between latitudes 27 -35° N and longitudes 120 – 128° E with varying domain resolutions from 1 –
100 km (Fig. 3).
This unstructured-grid high-resolution model has been developed to satisfy the computation requirements during storm surge
simulation, within the geographic coverage of the Shanghai sea and coastal area. This model system contains both the Shanghai
Coastal Typhoon Storm Surge Model (SC-TSSM) and the regional Hengsha Island Typhoon Storm Surge Model (HI-TSSM).
Multiple physical factors are included in this model system, such as typhoon events, open ocean currents, astronomical tides,
surface waves and freshwater discharge. Surface Water Modelling System (SMS) was used to generate mesh for this study
since it has a more effective grid generation function than MIKE, and it can refine a flexible mesh gradually which cannot be
achieved in MIKE.
In this model, the effect of different shapes of the sea wall in the storm surge model is small, therefore the shape of the sea
wall was assumed to be trapezoidal. The height of the sea wall along the Shanghai coastline has been set to 6.37 m relative to
mean sea level. The manning number was chosen as the bed resistance factor, and it was set to 80 $m^{1/3}$/s for ocean and 32 $m^{1/3}$/s
for land area. For wind forcing, the input wind profile was generated from the computed results of typhoon model, including
air pressure and U/V component of wind velocity that varied in time and domain. Since the Yangtze Estuary is included in the
SC-TSSM, the river's discharge should be taken into consideration as a source of freshwater. Based on previous work, the
discharge of Yangtze River has been set to 45 000 m³/s as the mean discharge for the period of July-September (Ge 2010).
As shown in Fig. 7, the results suggest that the SC-TSSM can simulate the propagation of storm surge satisfactorily. In general,
the numerical computation results are in good agreement with the observations, although some sections of the simulation are
under-predicted. For example, the differences between computation and observation are in the range of 0.2 – 0.5 m from 17 to
19 September 2007.

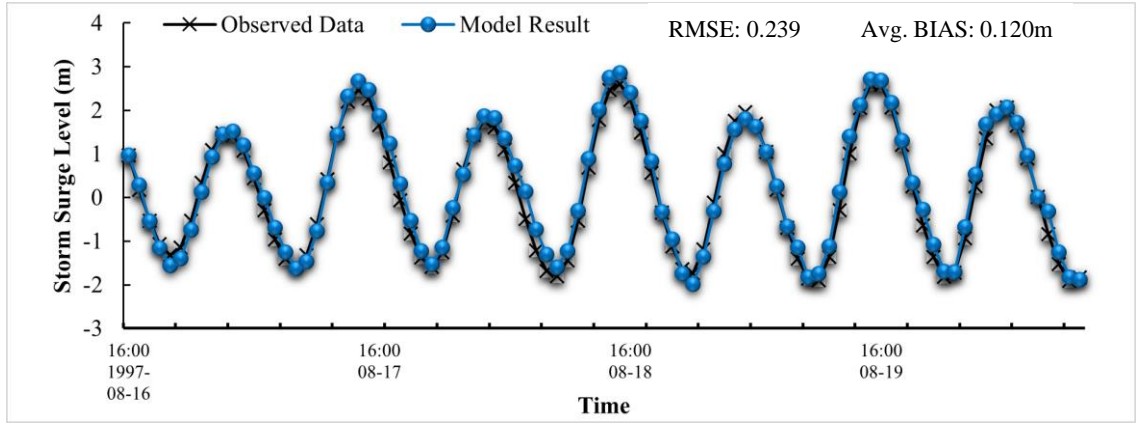

(a) Typhoon Winnie at Daji Station

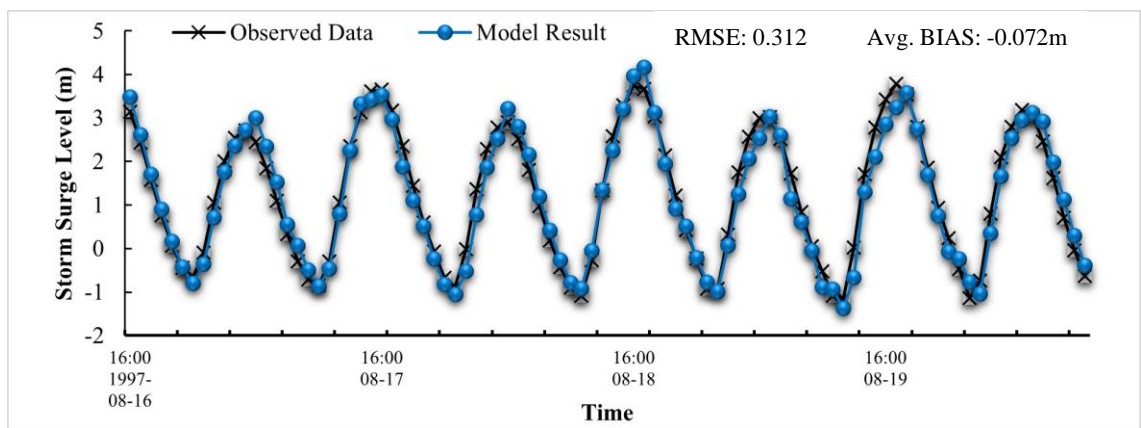

(b) Typhoon Winnie at Tanxu Station

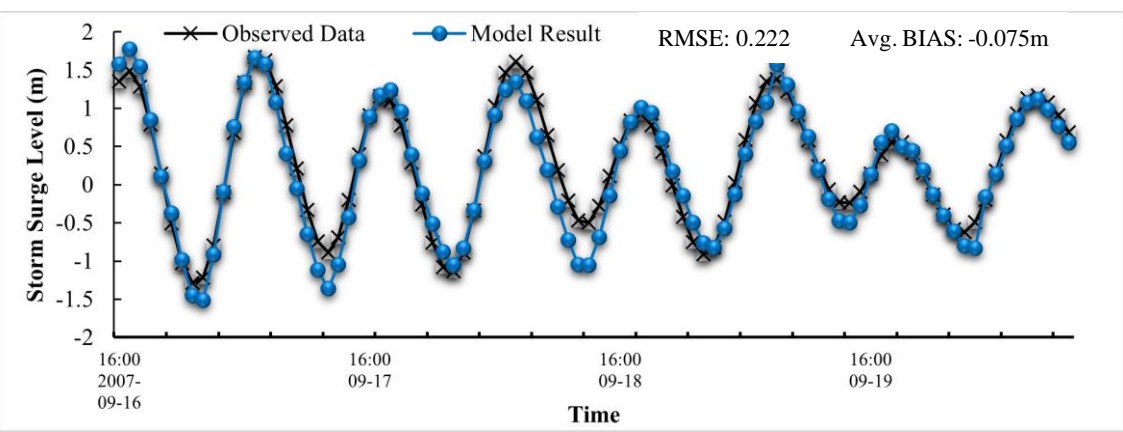

(c) Typhoon Wipha at Daji Station

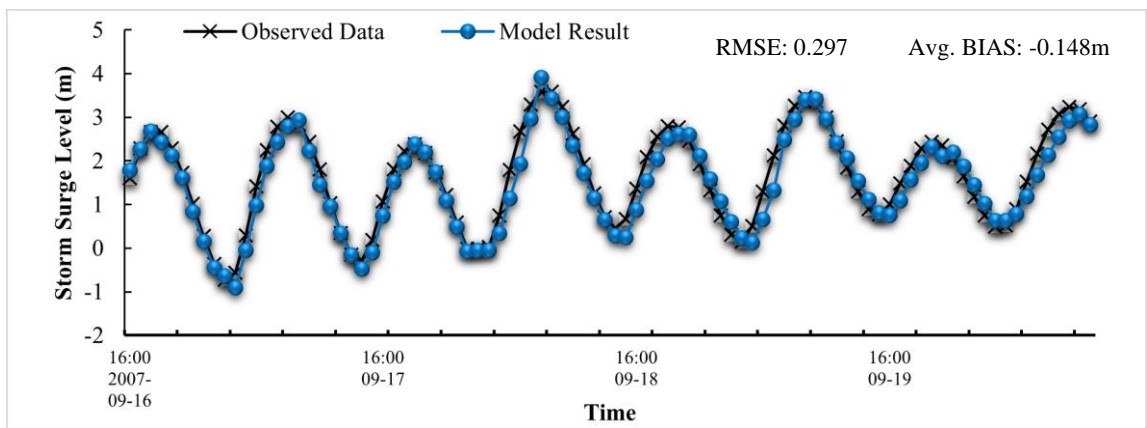

(d) Typhoon Wipha at Tanxu Station

**Fig. 7** Comparison of the observation data (black) and simulated results (blue) of storm surge levels (a) and (b) during Typhoon Winnie and (c) and (d) during Wipha at Daji and Tanxu stations.

Based on simulation results from MIKE 21, distribution maps of storm surge inundation and inundation depth in Shanghai, during the two case study typhoon events, are presented in Fig. 8. Simulation results show both typhoons gave rise to storm surge inundation in Shanghai across a large area. The distribution of storm surge inundation caused individually by Winnie and Wipha was basically the same but with a few differences in flood depth observed along the coastline and on the east and north coasts of Chongming and Hengsha Islands. The average inundation levels, of the storm surge that occurred during these two typhoons, were 1.78 m in 1997 (Winnie) and 0.9 m in 2007 (Wipha) in eastern Shanghai.

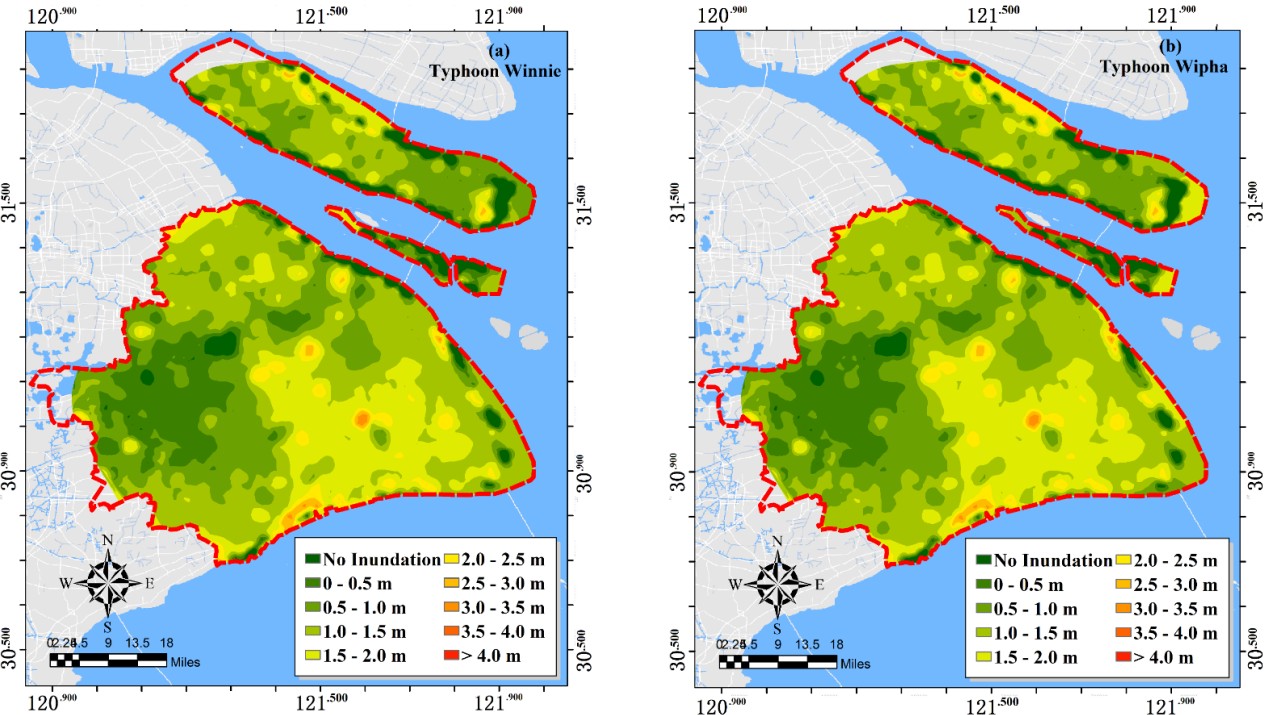

**Fig. 8** Distribution of the Maximum Inundation Area and Depth over Shanghai During (a) Typhoon Winnie and (b) Wipha

In order to analyse the effect of typhoon storm surge may have on reclamation projects around Hengsha Island, a survey line and six survey points along the south bank of the on-going project have been drawn (Fig. 9(a)). As the tide moved toward the south bank of Hengsha Island during the study period in the Yangtze Estuary, the time series of water level at these survey points can reveal the variation characteristics of storm surge in this area. The water level and wave speed at these six survey

points have been extracted from simulations in the HI-TSSM (Hengsha Island Tropical. An hourly output from the HI-TSSM
for the period from 18 hours before the landfall of the typhoon to 12 hours after demonstrates the differences of surge elevation
and speed between the different locations (Fig. 9(b) and (c)).

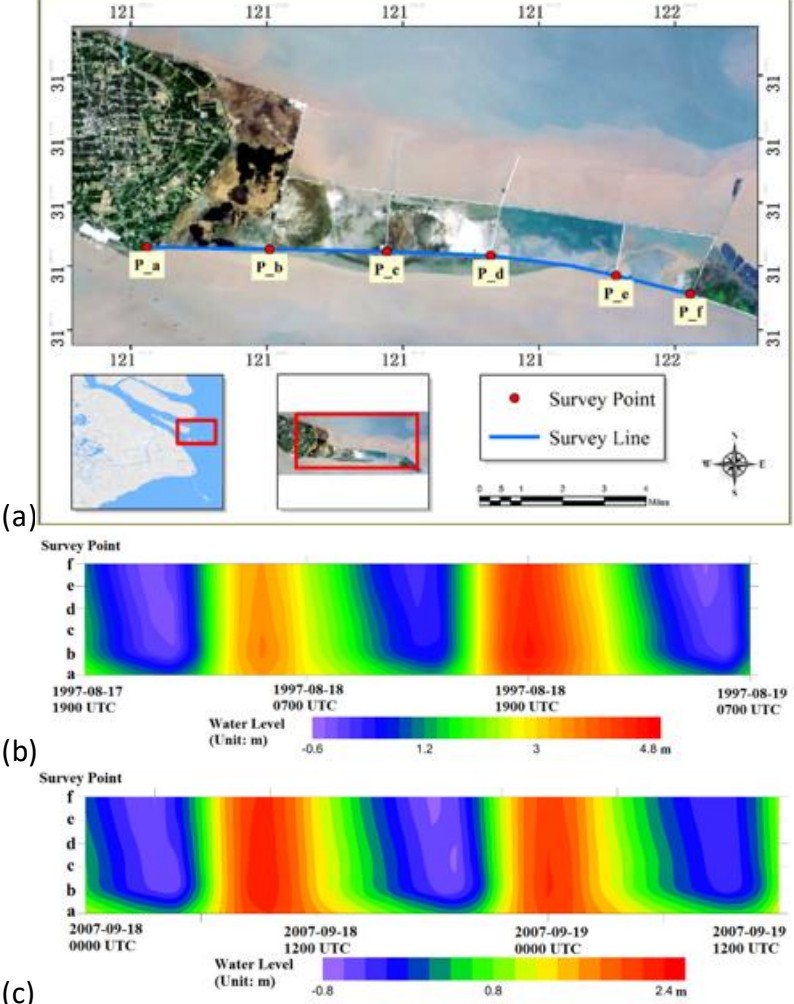


**Fig. 9** (a) Location of the survey line (blue) and six survey points (red) along the south bank of the reclamation project around
Hengsha Island, and distribution of water level (m) at the survey points during (b) Winnie and (c) Wipha. Sources: Esri,
DigitalGlobe, GeoEye, i-cubed, USDA FSA, USGS, AEX, Getmapping, Aerogrid, IGN, IGP, swisstopo, and the GIS User
Community.
Water levels at these survey points decreased slightly from points a to f. Differences in water level between these selected
points were larger during low tide than high tide. The difference between points a and f was 0.72 m (Winnie) and 0.52 m
(Wipha) at high tide, while it was up to 1.79 m and 1.32 m respectively during low tide (Fig. 8(c)). Coastal vulnerability to
typhoon storm surge inundation defined in this study is sensitive to the elevation of storm surge, especially during high tide.
Therefore, although there are only slightly differences in surge level (0.52 – 1.79 m between points a and f) along the survey
line, will lead to a variation of coastal inundation vulnerability to storm surge. The results also imply that vulnerability of land
reclamation to typhoon storm surge varies from place to place. Therefore, it is important to analyse coastal vulnerability to
storm surge inundation of reclaimed land before allocating different land use types. Better understanding of such vulnerability
will also provide crucial support to stakeholders for them to generate sustainable effective coastal protection strategies.

Generally, the mouth of Yangtze River, Hangzhou Bay, Chongming, and Hengsha Islands, and the riverbank along the Dazhi and Huangpu Rivers were the most seriously affected areas during these typhoon storm surge inundations. The inundation depths at these places were usually over 1.0 m. Maximum inundation depth in those areas reached 3.82 m during Winnie, and 2.65 m during Wipha. Severe storm surge led to widespread flooding, and the airport, factories, and warehouse, commercial and residential buildings were flooded. Combined with heavy rainfall, this meant the transportation system was disrupted, including communication lines and international airports. The detailed information on storm surge inundation in Shanghai during Winnie and Wipha were not available, thus the oceanic disaster communique of China published by State Oceanic Administration (1989-2015) was utilized to validate the inundation situation in Shanghai. The simulation results were in good agreement with the published descriptions, and in line with previous studies in Shanghai (Chen and Wang 2000; French 2001; Hu et al. 2005; Hu and Jin 2007; Ge 2010; Yin 2011; Yin et al.,2013a; Harwood et al. 2014).

The results from this study also suggested that the height of storm surge along the Huangpu River and Dazhi River basin was high and the riverbanks experienced serious flooding during typhoon induced storm surge, which was also reported by the oceanic disaster communique of China published by State Oceanic Administration (1989-2015). Previous studies failed to capture these features (Chen and Wang 2000; French 2001; Hu et al. 2005; Hu and Jin 2007; Ge 2010; Yin 2011; Yin et al. 2013a; Harwood et al. 2014).

**5 Discussion**

For typhoon storm surge modelling in this study, we demonstrate that a meso-scale simulation can be used to compute storm surge inundation and assess the inundation vulnerability of different land use types. This study enlarges the body of knowledge on storm surge studies in Shanghai, and also proposes a meso-scale simulation can be used for coastal planning purposes. Previous studies of storm surge were usually conducted at national or local levels (Butler et al. 2012; Dietrich et al. 2011a). In China, most of these studies tended to emphasize the significance of numerical modelling of storm surge and risk analysis either for the coastline on a large spatial scale (>100 km in length) (Tan et al. 2011; Yin 2011; Zheng 2010) or for the small scale coastal area (1 – 1000m in length) with fine resolution simulation (Xie 2010; Xie et al. 2010; Ye 2011; Zhang et al. 2006). The majority of these studies concentrated on three districts in the Shanghai coastal area, namely, the Pudong, Jinshan, and Fengxian Districts (Xie 2010; Ye 2011). These studies probably needed to pay more attention to the river basins. However, results from this study show that the river basins of the Dazhi and Huangpu Rivers were among the most serious impacted areas during Typhoons Winnie and Wipha. Meso-scale (1 – 100 km in length) studies on storm surge have not been conducted for Shanghai, and the meso-scale framework in this study fills the gap.

Large and small-scale simulations do each have their own advantages. For example, large-scale simulations require low consumption of computation resources and time depending on the resolution used. Large-scale studies therefore could be applied on a national scale to analyse typhoon storm surge impacts, to simulate typhoons and storm surge changes over time, and to provide necessary data to propose general plans for hazard mitigation. Small-scale simulations usually involve fine spatial resolutions, ranging from 5 m to 100 m, in order to capture subtle changes of the flood waters. Nonetheless, neither large nor small-scale simulations always fit for coastal planning purposes. Large-scale simulation is not suitable for local planning because its coarse spatial resolution cannot reflect the detailed distribution of storm surge inundation. Although numerical simulation, in the context of coastal planning, requires a significant number of accurate and detailed computation results at a regional level, high spatial resolution at the local scale will have high costs in terms of computation resource and time. For example, a small-scale model with a fine spatial resolution mesh of 100 m – 1 km was initially used in this study, covering only the estuary and coastal area. It required over 600 hours to run one simulation on a computer with 16G RAM, 500G SSD, quad-core Intel Core i5 processors. Compared to the 600 hours of computation time by small scale model, the

multi-nested meso-scale model only required about 30 hours to run a single simulation with a reasonable accuracy where required. Meso-scale studies could therefore not only fulfil the requirements for simulation accuracy, but also take less time and resource. They are more suitable for use when a large number of simulations are required over a long-time scale. By implementing this meso-scale model, the focus of storm surge simulation is at an appropriate medium scale to fit planning purposes.

The simulations conducted in this study has enlarged the body of knowledge about storm surge inundation in Shanghai and suggested that more attention needs to be paid not only to the area along the coastline, but also to the nearby rivers. Some studies in Shanghai started to look at the inundation along Huangpu River caused by typhoon storm surge (Borsje et al. 2011; Yin et al. 2013a). Globally, the work conducted by Rupp and Nicholls (2002) on the river Thames emphasized the interaction of surge and tide in river basin. Ali (1996) also demonstrated in their study that the most severe inundation area during a synthetic typhoon in eastern North Carolina was in the Pamlico River region.

**6 Conclusions**

This paper developed a resource and time efficient approach for simulating typhoon-generated storm surge, which can be applied to coastal mega-cities around the world, even where flood observation data is inadequate. Typhoon induced storm surge was simulated in Shanghai and inundation maps were drawn in ArcGIS. These maps provide a clearer picture of the spatial distribution and the variation of such vulnerability over Shanghai. Results showed the south of Shanghai, the riverbanks along the Huangpu and Dazhi Rivers and most of Chongming Island were subject to serious typhoon storm surge inundation. It also showed that reclamation land, such as that on Hengsha Island is particularly vunerable to storm surge innundation. The meso-scale simulation method proposed in this study provides a realistic storm-surge innundation result at the city level. Furthermore, due to its low data and time consumption, this approach can be implemented when a large number of models are required for mitigation and planning.

**Author Contribution**

Dong, Stephenson and Wakes devised the numerical experiments. Dong carried out the numerical simulations and analysis of the results. Chen and Ge supplied the validation data, commented and advised on model construction and outputs. Dong, Stephenson and Wakes prepared the manuscript.

**Acknowledgements**

The authors would like to acknowledge the University of Otago PhD scholarship that funded this work and DHI for access to MIKE 21.

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

570    **Table 1** Major configuring parameters for the simulation models

| Model Parameter | Configuration |
| --- | --- |
| Minimum Time Step | 0.01 sec |
| Maximum Time | 30 secs |
| Critical CFL Number | 0.8 |
| Drying Depth | 0.005 m |
| Flooding Depth | 0.05 m |
| Wetting Depth | 0.1 m |
| Manning Number | 80 $m^{1/3}$/s for ocean, 32 $m^{1/3}$/s for land |
| Neutral Pressure of Wind Field | 1008 hPa |
| Soft Start Interval for Wind | 3600 secs |
| Freshwater Discharge | Simple Source, 45 000 $m^3$/s |

571

| Model Parameter | Configuration |
| --- | --- |