# Peer review of "Meso-scale Simulation of Typhoon-Generated Storm Surge"

_Natural Hazards and Earth System Sciences, 2017_

## Referee Comment (RC1) · Anonymous Referee #1 · 1 Jun 2017

This manuscript investigates the storm surge simulation using meso-scale regional model. A series of numerical simulations showed that reliable result. However, further more detailed analysis is still needed in this manuscript. The manuscript is not generally well written and the figures are not well constructed and clear to understand. In addition, there is no scientific distribution compared with recent storm surge research. The research purpose is not clear. There is nothing of new finding or argument in this paper. Furthermore author didn't show simulation result clearly. There are some issues that need to be resolved before this manuscript can be acceptable for publication, as summarized below.

Major concerns

1. Meso-scale simulations have some advantage for the numerical study. You need

to describe why you choose this method. Because you also used fine grid resolution(under the 1km resolution grid) when using nesting simulation. 2. When you conduct numerical study, you should present the description of model and detailed model setup information. The detail description about numerical model setup is positively necessary. Model description & introduction, Grid information(grid size and shape, . . .), bottom drag coefficient, boundary condition, forcing generation information(tidal force, surface forcing, . . .). This is basic part for the numerical model research. 3. As an explanation of storm surge model in chapter 3, description about typhoon wind model is recommended. Sample plot of wind & pressure field generation is recommended. In addition I wonder how does the typhoon field move for each 6 hours interval? 4. You need to explain more about blending of simulated typhoon wind field and ECMWF dataset. 5. Furthermore, You need to explain why you choose typhoons Winnie & Wipha. You suggested just two observation station data. I think there are much more tidal station and wave station data available. 6. You need to classify water level data to tide and storm surge component. After finishing well simulation of tide component, you can suggest comparison of surge simulation result. 7. Generally more figures are needed to explain the simulation result. 8. There is no figure and information for inundation modelling. You mentioned Shanghai coastline has been set to 6.37m above MSL. In this study, simulated maximum water level was recorded under the 5m from Figure 3. 9. I cannot understand horizontal axis description in the Figure 3. What time does that mean? And I cannot find surge height clearly in this figure. 10. You need to suggest observed inundation trace map with figure 4. The figure 4 shows just result of simulation 11. I don't understand what you want to say in this research. If the purpose of this research is introduction of meso-scale modelling's advantage, this conclusion does not have scientific distributions. 12. You mentioned this research shows successful agreement of storm surge simulation. However, you need to explain further more description of simulation result. 13. In addition I recommend you suggest spatial distribution of the storm surge and wave distributions. You still does not suggest any meaningful analysis from this figure 3,4.

Other concerns

1. A lot of sentences of manuscript need corrections by a native English speaker. 2. Many part of previous work and model description are repeated in the article. 3. Some sentences in conclusion are repeated at the abstract. Use other expresstion.

Please also note the supplement to this comment:
http://www.nat-hazards-earth-syst-sci-discuss.net/nhess-2017-34/nhess-2017-34-RC1-supplement.pdf

---

## Referee Comment (RC2) · Anonymous Referee #2 · 6 Jun 2017

This paper details a hydrodynamic modelling study of the storm surge inundation produced by two historical typhoons (Winnie in 1997and Wipha in 2007) in Shanghai. A hydrodynamic model was set up using an unstructured grid mesh varying from 100 km offshore to 100 m near to the shore then a high resolution coastal model based on MIKE 21 was used to simulate the overland flooding. The authors claim that the novelty in their modelling approach is the use of the two models to ensure maximum resolution of results at the coast without exceptionally high computer overheads. The results consist of time series validation at two tide gauges and an inundation map for Shanghai for each event, which is claimed to align well with observed inundation. The authors go on to conclude that their model results based on MIKE (a commercial software product) show that the MIKE software is competitive with other open-sourced codes such as

[Figure]

ADCIRC and FVCOM. In general, I was not convinced by the arguments put forward by the authors. Nested modelling strategies to maximise coastal details while employing larger regional scale models to provide the boundary conditions have been around for decades. I found this paper to provide little new insight into the modelling of storm surge inundation. The structure of the paper was confusing. The details of the models are discussed throughout the paper making it difficult for the reader to seek the details of the model setup from what was reviews of other studies. For example, there is some description of the modelling approach in section 3.2 and also in 4.3, which should be the results section. The results section was disappointingly slim. It consisted only of a time series comparison at two tide gauges (remarkably in agreement with the observed sea levels during each event) and a single inundation map of maximum inundation area and water depth. How much of the total water level was storm surge and how much was tides in each event? What about validation of the typhoon winds that the authors note is key to accurately simulate the storm surges? It is not possible to tell when exactly the typhoon occurred from the water level time series. A map of the cyclone tracks and the tide-only contribution and the residual difference would be useful. The authors claim that other studies have failed to pay enough attention to the river basins whereas they claim the resolution of the model in the present study fills this gap. However, the authors do not detail how they have modelled the rivers. How is the input from the rivers incorporated into their model grid? There is no mention of including flow hydrographs as boundary conditions for the terrestrial input. Is rainfall flooding a contributing factor in addition to the storm surge from the sea? What about other factors that contribute to total water levels such as from wind waves (setup and runup)? My conclusion is that this paper requires considerable work to make it acceptable for publication. If its purpose, as the authors claim, is to provide a new modelling strategy that addresses a resolution gap not addressed by previous studies, then they need to more carefully document what other studies have done and show how their study addresses this gap. They also need to provide more details of their model setup and the processes that they have accounted for and those they have neglected and discuss the significance of

these. Finally the organisation of the paper needs to be clearer. I therefore recommend rejection of this paper in its current form.

I offer some more specific comments below title There should be a hyphen between 'typhoon' and 'generated' Abstract A sustainable urban plan relies on sound preparedness... (i.e replace 'well' with 'sound') Page 2, line 15 Insert 'the' in '...three types based on the scale of modelling...' In the first paragraph of section 2, the authors discuss large, meso and small scale studies. Many references are quite old now. There have been many more studies undertaken post-2012. Also the way in which some of these studies is described is not strictly correct. E.g. the authors describe McInnes et al as being a large-scale study. It focused on a small regional town in Australia, although it did use a nested approach to achieve a similar goal to the study presented here. More recent studies in Australia include for example those of Haigh et al 2014 a, b or McInnes et al, 2013.

references: Haigh I, Wijeratne EMS, MacPherson L, Pattiaratchi C, Mason M, Crompton R, George S (2014a) Estimating present day extreme water level exceedance probabilities around the coastline of Australia: tides, extratropical storm surges and mean sea level. Clim Dyn 42:121–138. doi:10.1007/s00382-012-1652-1 Haigh I, MacPherson L, Mason M, Wijeratne EMS, Pattiaratchi C, Crompton R, George S (2014b) Estimating present day extreme water level exceedance probabilities around the coastline of Australia: tropical cyclone induced storm surges. Clim Dyn 42:139–147. doi:10.1007/s00382-012-1653-0 McInnes KL, Macadam I, Hubbert G, O'Grady J (2013) An assessment of current and future vulnerability to coastal inundation due to sea-level extremes in Victoria, southeast Australia. Int J Climatol 33:33–47. doi:10. 1002/joc.3405

Page 3 line 17 Change to '..pressure fields were calculated...' Page 3 line 18 Change to '...collected to validate the hydrodynamic models.' Pate 3 line 22 Suggest to use the word surge not wave here so as not to confuse with wind-generated waves Page 4 line 7 Change to 'tide constituents are prepared' Page 4 line 12 Change to '.. to provide

accurate wind and pressure..' Page 5 line 13 Insert 'model' after 'hydrodynamic' Page 5 line 14 Change to 'typhoon-induced' Page 6 line 28 '. . . regarded as real. . .' . I suggest changing this to 'regarded as providing a close approximation of the state of the atmosphere'. Page 6 line 28 If the ECMWF data is such a good approximation, then why not dispense with the Holland vortex model all together? Page 7 line 14 Change to 'Shanghai lies at the half way point.' Page 9 line 14 'Simulated results have been passed to the storm surge model to generate wind-induced waves'?? Normally a storm surge model is a hydrodynamic model, incapable of simulating wind-waves. Can you clarify what is meant here? Page 9 Line 19 importance -> important Page 11 line 9 Mode -> model

---

## Author Comment (AC1) · 10 Dec 2019

1. Anonymous Referee #1 1.1 Meso-scale simulations have some advantage for the numerical study. You need to describe why you choose this method.

Reasons for choosing meso-scale method were explained in the introduction. We have added another statement in the second paragraph that justifies our approach with reference to the argument made by Ogie et al. (2019) for the need of less resource and data intense approaches to flood modelling in mega-cities.

1.2 The detail description about numerical model setup is positively necessary.

A new section (3.2.4) has been added describing the major model parameters, including density, horizontal eddy viscosity, Coriolis forcing, tidal potential, sea wall and

boundary conditions, and other parameters. We have also added a Table that shows they model parameters to illustrate how the model was configured.

1.3 As an explanation of storm surge model in chapter 3, description about typhoon wind model is recommended. Sample plot of wind & pressure field generation is recommended. In addition, I wonder how does the typhoon field move for each 6 hours interval?

As for the typhoon field, the input data, best track data from CMA Tropical Cyclone Centre, were recorded every 6 hours. However, the wind and pressure field was calculated at 1 hour intervals. We have added this to section 3.4 lines 237-238.

1.4 You need to explain more about blending of simulated typhoon wind field and ECMWF dataset.

Reasons for blending of simulated wind field and ECMWF data was explained in the last paragraph on Page 9, and the approach and formula of such blending has been described in Page 10.

We have added a new figure that illustrates comparison of the two models against observed data (New Fig 1) and this shows how much better the Holland model is compared to the Young and Sobey.

1.5 Furthermore, you need to explain why you choose typhoons Winnie & Wipha. You suggested just two observation station data. I think there are much more tidal station and wave station data available.

Reasons for choosing Winnie and Wipha have already been explained on Page 11 and 12. There are more tide stations that recorded storm surge level during Winnie and Wipha than for other typhoon and provide the best data to calibrate simulation results. Other stations are either too far away from typhoon tracks or data were not available at the time we collected data.

1.6 You need to classify water level data to tide and storm surge component. After

Interactive
comment
finishing well simulation of tide component, you can suggest comparison of surge simulation result.

The model was validated based on the other two tide gauge stations, Location of these two stations is updated in fig 2. Tide simulation results are added on page 15-17. Comparison of surge simulation result was discussed on page 19-20.

1.7 Generally, more figures are needed to explain the simulation result.

We have added 5 new figures providing additional details of the modelling outputs. Fig 1 is added to show simulated results from Holland model and Young & Sobey model. Based on comparison in our case study, Holland model show that the model is in good agreement with the actual observation. Figure 2 shows typhoon tracks relative to Shanghai. Fig 4 is added to show the simulation results of tide component. We have added an enlarged image of the mesh around Shanghai in Figure 3. Fig 5 and 6 are added to show the output from the Holland model at 1 hr intervals and compared against observed data in terms of wind speed and direction. More results from HI-TSSM are added from page 23-24 to demonstrate the simulated results along the south bank of the reclamation project around Hengsha Island. Fig 9 is added to show the location of the survey line and six survey points along the south bank, and distribution of water level at the survey points during two selected typhoon.

1.8 There is no figure and information for inundation modelling. You mentioned Shanghai coastline has been set to 6.37m above MSL. In this study, simulated maximum water level was recorded under the 5m from Figure 3.

The key point is that the tide gauge stations are not on the sea wall. But our simulated max water level at the tide stations is in line with the observed data. Even though the observed water level was under 5m, Shanghai still suffered inundation. That means surge height increased between the tide stations and sea enough to overtop the wall. Our Figure 7 is showing our modelling has simulated surge well when compared to observed data at the stations.

1.9 I cannot understand horizontal axis description in the Figure 3. What time does that mean? And I cannot find surge height clearly in this figure.

Axis are time stamp from the simulation – New figure 7 reformatted

1.10 You need to suggest observed inundation trace map with figure 4. The figure 4 shows just result of simulation.

Observed inundation trace maps for the selected typhoon events are not available. Inundation modelling is validated based on the observed data. We stress that our model is designed to be used in city planning. Although there are a lot of meteorological and hydrological models that could be used for forecasting and hindcasting, they are not suitable for planning purpose due to the computational power required and insufficient data.

1.11 If the purpose of this research is introduction of meso-scale modelling's advantage, this conclusion does not have scientific distributions.

We have clarified the conclusion with a focus on advantages of meso-scale modelling, deleted other un-related parts.

1.12 I recommend you suggest spatial distribution of the storm surge and wave distributions. You still do not suggest any meaningful analysis from this figure 3,4.

Further description on simulation results has been provided on Page 23-25.

1.13 A lot of sentences of manuscript need corrections by a native English speaker.

Quality of all the figures have been improved and further proofreading will be done before publishing.

1.14 Many part of previous work and model description are repeated in the article.

Descriptions on previous work and model in this paper have been simplified, with repetition removed.

[Figure]

1.15 Some sentences in conclusion are repeated at the abstract. Use other expression.

Changed the expressions in conclusion, and revisited the purpose of this paper.

---

## Author Comment (AC2) · 10 Dec 2019

Anonymous Referee #2 It consisted only of a time series comparison at two tide gauges (remarkably in agreement with the observed sea levels during each event) and a single inundation map of maximum inundation area and water depth. How much of the total water level was storm surge and how much was tides in each event?What about validation of the typhoon winds that the authors note is key to accurately simulate the storm surges?

It is not possible to tell when exactly the typhoon occurred from the water level time series. A map of the cyclone tracks and the tide-only contribution and the residual difference would be useful.

RESPONSE: Figure 2 has been changed to include typhoon tracks. As we have added 4 new figures to provide additional modelling details (and this is a method focused paper) we feel more figure is unnecessary. The text (page 13) already contains details of the timing and extent of both typhoon.

The authors claim that other studies have failed to pay enough attention to the river basins whereas they claim the resolution of the model in the present study fills this gap.

RESPONSE: Here we are commenting of the absence of considering storm surge flooding into the river mouths, rather than the discharge of water from river basins. Our modelling showed this is important in the case of Shanghai and had not previously been recognized.

However, the authors do not detail how they have modelled the rivers. How is the input from the rivers incorporated into their model grid?

RESPONSE: Model grid is only designed for the coastal area, rivers are not modelled in this study, freshwater discharged is considered and setup in the model (refer to Table 1)

There is no mention of including flow hydrographs as boundary conditions for the terrestrial input. Is rainfall flooding a contributing factor in addition to the storm surge from the sea? What about other factors that contribute to total water levels such as from wind waves (setup and runup)?

RESPONSE: Our modelling doesn't include these variables in order to operate at the scale we are focused on.

2.1 Title Changed 'typhoon generated' to 'typhoon-generated'. 2.2 Abstract, Page 1, line 11 Changed 'sustainable urban plan relies on well preparedness' to 'sustainable urban plan relies on sound preparedness'. 2.3 Page 2, line 15 Changed to '...the three types based on the scale of modelling...'. 2.4 In the first paragraph of section 2,

many references are quite old now. Corrected and updated the reference used here. 2.5 Page 3, line 17 Changed to '...pressure fields were calculated...'. 2.6 Page 3, line 18 Changed to '...collected to validate the hydrodynamic models...'. 2.7 Page 3, line 22 Changed to 'wind-generated surges'. 2.8 Page 4, line 7 Changed to 'tide constituents are prepared'. 2.9 Page 4, line 12 Changed to '... to provide accurate wind and pressure...'. 2.10 Page 5, line 13 Changed to 'hydrodynamic model'. 2.11 Page 5, line 14 Changed to 'typhoon-induced'. 2.12 Page 6, line 28 Changed '... regarded as real...' to 'regarded as providing a close approximation of the state of the atmosphere'. 2.13 Page 6, line 28 If the ECMWF data is such a good approximation, then why not dispense with the Holland vortex model all together?

ECMWF reanalysis dataset has a good spatial resolution of 0.25 °. On the one hand, the quality of ECMWF is not as good as simulated results from the local typhoon best track data. On the other hand, the simulation results from the Holland model have shown that the wind speed after the typhoon makes landfall was much lower than the measured data. To improve the quality of typhoon simulated results, these two data have been blended.

2.14 Page 7, line 14 Changed to 'Shanghai lies at the half way point...'. 2.15 Page 9, line 14 'Simulated results have been passed to the storm surge model to generate wind-induced waves'?? Normally a storm surge model is a hydrodynamic model, incapable of simulating wind-waves. Can you clarify what is meant here? Changed this sentence to 'Computed results from the typhoon model have been passed to the storm surge model to simulate typhoon-generated storm surge.'. 2.16 Page 9, Line 19 Changed 'importance' to 'important'. 2.17 Page 11, line 9 Changed 'Mode' to 'model'.

---

## Author Response (AR3)

We thank the reviewer for their careful attention o detail and have undertaken both major recommendations.

>1. In the copy of the paper sent to me Figures 1, 4, 5, 6 and 7 all contain parts of the material plotted that lie outside the boundaries of the figure.

Figures have been fixed to display correctly. This appears to be a problem converting Excel files embed in word to pdf format. We have removed these from the word document and replaced with picture formatted images. These now display correctly in the pd. We apologies for not noticing this error at our last submission.

2. The authors use "typhoon" as a plural word. This is incorrect.

We have edited all places where typhoon should be plural.